# Structural and functional characterization of AfsR, an SARP family transcriptional activator of antibiotic biosynthesis in *Streptomyces*

Yiqun Wang[1], Xu Yang[1], Feng Yu[1], Zixin Deng[1], Shuangjun Lin[1], Jianting Zheng[1,2]*

1 State Key Laboratory of Microbial Metabolism, School of Life Sciences and Biotechnology, Shanghai Jiao Tong University, Shanghai, China, 2 Joint International Research Laboratory of Metabolic & Developmental Sciences, Shanghai Jiao Tong University, Shanghai, China

* jtzheng@sjtu.edu.cn

**Data Availability Statement:** Atomic coordinate has been deposited in PDB with accession numbers 8HVR (SARP-TIC) and 8JKE

## Abstract

*Streptomyces* antibiotic regulatory proteins (SARPs) are widely distributed activators of antibiotic biosynthesis. *Streptomyces coelicolor* AfsR is an SARP regulator with an additional nucleotide-binding oligomerization domain (NOD) and a tetratricopeptide repeat (TPR) domain. Here, we present cryo-electron microscopy (cryo-EM) structures and in vitro assays to demonstrate how the SARP domain activates transcription and how it is modulated by NOD and TPR domains. The structures of transcription initiation complexes (TICs) show that the SARP domain forms a side-by-side dimer to simultaneously engage the *afs box* overlapping the −35 element and the σ$^{HrdB}$ region 4 (R4), resembling a sigma adaptation mechanism. The SARP extensively interacts with the subunits of the RNA polymerase (RNAP) core enzyme including the β-flap tip helix (FTH), the β′ zinc-binding domain (ZBD), and the highly flexible C-terminal domain of the α subunit (αCTD). Transcription assays of full-length AfsR and truncated proteins reveal the inhibitory effect of NOD and TPR on SARP transcription activation, which can be eliminated by ATP binding. In vitro phosphorylation hardly affects transcription activation of AfsR, but counteracts the disinhibition of ATP binding. Overall, our results present a detailed molecular view of how AfsR serves to activate transcription.

## 1. Introduction

Streptomycetes are multicellular bacteria with a complex developmental cycle [1]. They produce numerous bioactive natural products, including many antibiotics with important applications in medicine and agriculture. This capability makes streptomycetes one of the most important industrial microbial genera. Complex regulatory systems tightly govern the natural product biosynthesis of streptomycetes, involving both global and pathway-specific regulators. *Streptomyces* antibiotic regulatory protein (SARP) family regulators include many of the pathway-specific regulators and some global regulators. They are prominent initiators and activators of natural product biosynthesis and have mainly been found in actinomycetes [2].

(AfsRT337A-TIC). Four cryo-EM density maps have been deposited in Electron Microscopy Data Bank with accession number EMD-35047 (SARP-TIC), EMD-36370 (AfsRT337A-TIC), EMD-35046 (SARP region from SARP-TIC) and EMD-36369 (SARP region from AfsRT337A-TIC). All data underlying the main and supplementary figures can be found at supplementary S1 Data and S1 Raw Images.

**Funding:** This work was supported by National Key R&D Program of China (2019YFA0905400) and National Natural Science Foundation of China (32070040, 32370071). The funders had no role in study design, data collection and analysis, decision to publish, or preparation of the manuscript. URLs: https://chinainnovationfunding.eu/national-key-rd-programmes/; https://www.nsfc.gov.cn/.

**Competing interests:** The authors have declared that no competing interests exist.

**Abbreviations:** BTA, bacterial transcriptional activation; CAP, catabolite activator protein; CRP, cAMP receptor protein; cryo-EM, cryo-electron microscopy; DR, direct repeat; dsDNA, double-stranded DNA; FDR, false discovery rate; FHA, forkhead-associated; FTH, flap tip helix; HTH, helix-turn-helix; MS, mass spectrometry; NBD, nucleotide-binding domain; NOD, nucleotide-binding oligomerization domain; ODB, OmpR-type DNA-binding; RNAP, RNA polymerase; SARP, *Streptomyces* antibiotic regulatory protein; SEC, size-exclusion chromatography; STAND, signal transduction ATPases with numerous domains; TIC, transcription initiation complex; TPR, tetratricopeptide repeat; TSS, transcription start site; ZBD, zinc-binding domain.

Recently reported members of the SARP family include ChlF2 from *Streptomyces antibioticus* and MilR3 from *Streptomyces bingchenggensis*. ChlF2 activates the production of chlorothricin, known for its anti-inflammatory properties, and MilR3 stimulates the synthesis of an excellent insecticide milbemycin [3,4]. SARPs feature an N-terminal OmpR-type DNA-binding (ODB) domain and a C-terminal bacterial transcriptional activation (BTA) domain, collectively referred to as the SARP domain. The ODB domain is characterized by a winged helix-turn-helix (HTH) comprising 3 α-helices and 2 antiparallel β-sheets [5], whereas the BTA domain is all-helical [6]. In contrast to "small" SARPs that only contain an SARP domain, "large" SARPs possess extra domains at their C-terminals that are hypothesized to modulate the activity of the SARP domain (Fig 1A). "Small" SARPs are more frequently observed and extensively studied among actinomycetes, compared to their "large" counterparts [7]. SARPs bind to the target promoters having common features. They contain direct repeats, the 3′ repeat of which is located 8 bp from the −10 element, and each repeat is separated from the adjacent repeat by 11 bp or 22 bp, corresponding to 1 or 2 complete turns of the DNA helix, respectively [8].

*Streptomyces coelicolor* AfsR is a pleiotropic, global regulator of both primary and secondary metabolism belonging to the "large" SARP group, containing an SARP (Met1-Ala270) domain, a conserved approximately 35 kDa dubbed nucleotide-binding oligomerization domain (NOD) (Ala271 to Glu618), an arm domain (Arg619-Glu777) as well as a tentative tetratrico-peptide repeats (TPRs) sensor domain (Asp778-Arg993) [9] (Fig 1A). AfsR is found for its vital role in antibiotic biosynthesis. AfsR null mutants show defects in *afsS* transcription, as well as production deficiencies in actinorhodin and undecylprodigiosin [10]. It is presumed that upon binding of AfsR to a 22-base pair (bp) binding box (*afs box*), RNA polymerase (RNAP) holoenzyme is recruited to the *afsS* promoter, allowing for transcriptional initiation [9]. AfsS serves as a master regulator of secondary metabolism and nutritional stress response [11]. AfsR-AfsS system is widely distributed among *Streptomyces*, and expressing AfsR in the heterologous hosts can awaken silent antibiotic production genes [12]. Additionally, AfsR and the master regulator PhoP bind to overlapping sequences within PhoR-PhoP regulon promoters, such as *pstS*, *phoRP*, and *glnR*, exerting crosstalking regulatory control over the response to phosphate and nitrogen scarcity [13,14] (Fig 1B). AfsR can be phosphorylated by several serine/threonine kinases including AfsK, AfsL, and PkaG, and achieves better DNA binding affinity [12,15]. The full-length AfsR, as well as truncations containing only the SARP domain or lacking the TPR domain, can bind to *afs box* and activate *afsS* transcription in vitro [9]. However, the NOD is essential for the functionality of AfsR in vivo [10].

The RNAP holoenzyme consists of a dissociable σ subunit that establishes specific interactions with the −35 and −10 elements of the promoter to initiate transcription. The positioning of RNAP is primarily influenced by the contacts between σ region 4 (R4) and the −35 element, while the formation of an open complex is driven by the contacts between σ region 2 (R2) and the −10 element [16,17]. The *afsS* promoter comprises a suboptimal −10 element (CACTGT) and a poor −35 element (TTCAGC). This characteristic is commonly observed in streptomycete promoters [18,19], which complicates the identification of the promoters. The 22-bp *afs box*, centered at position −29.5 relative to the transcription start site (TSS) in the *afsS* promoter, overlaps with the spacer between the −10 and −35 elements. It is composed of two 11-bp direct repeats (DRs), with the upstream DR (−40 to −30) overlapping with the −35 element (−38 to −33) [20,21]. The catabolite activator protein (CAP), also known as the cAMP receptor protein (CRP), is an extensively studied transcription factor. The 22-bp CAP binding box (*cap box*) is centered at −61.5 in class I promoters and at −41.5 in class II promoters [22–24]. The binding position of AfsR suggests that it may possess a distinct activation mechanism.

Due to the presence of the NOD domain, AfsR is also classified as the signal transduction ATPases with numerous domains (STAND) family [25]. Representative examples of STAND

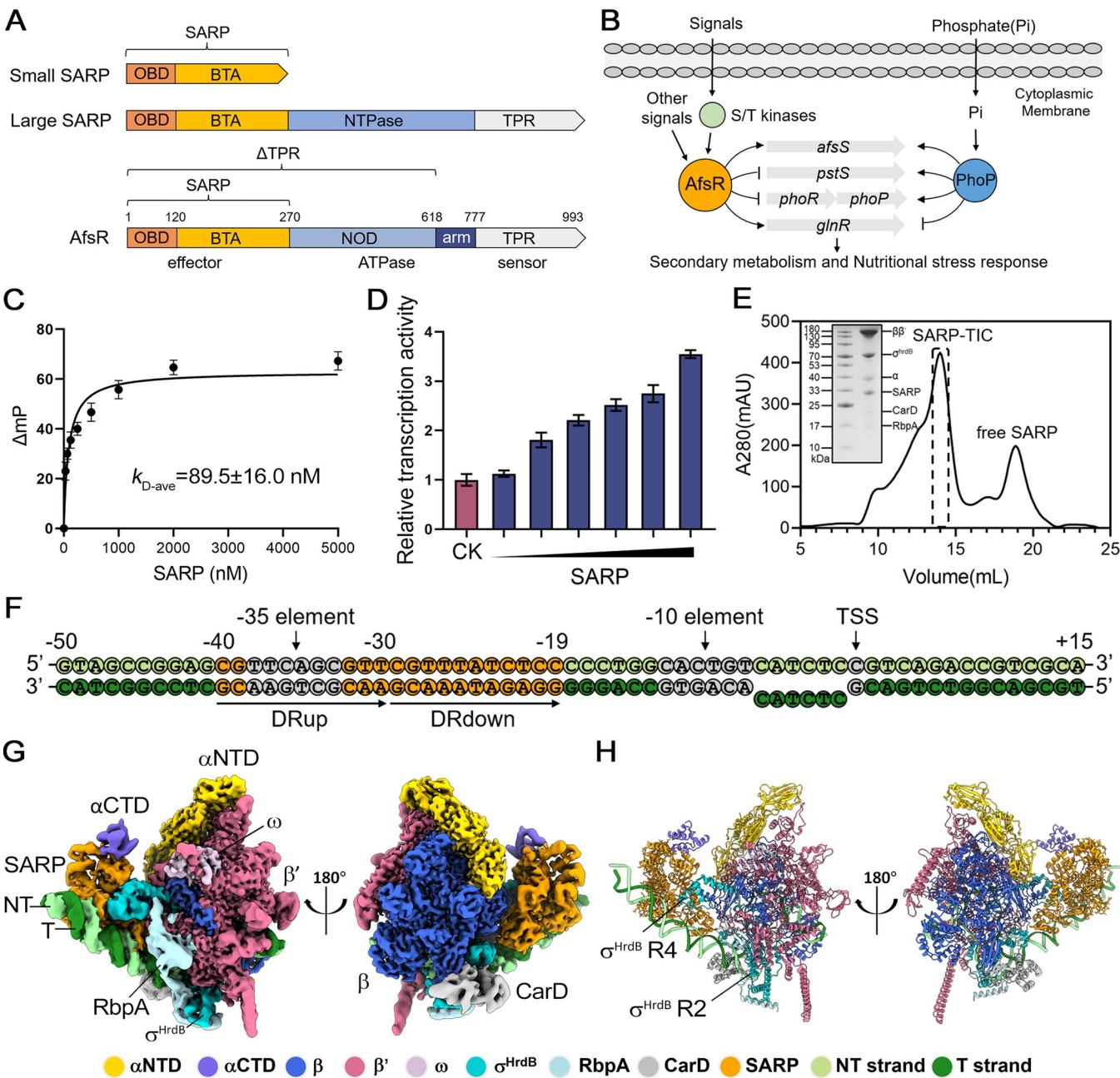

**Fig 1. The TIC of the SARP, RNAP, and *afsS* promoter DNA.** (A) Domain structures of SARPs. The N-terminal ODB together with the following BTA is referred as an SARP domain. *S. coelicolor* AfsR is a large SARP with additional NOD and TPR domains. (B) Hypothetical scheme for the regulation of *S. coelicolor* AfsR. (C) Fluorescence polarization assays of the SARP with *afs box*. Error bars represent mean ± SEM of *n* = 3 experiments. (D) Transcription assays with increasing concentrations (62.5 nM, 125 nM, 250 nM, 500 nM, 750 nM, 1,000 nM) of the SARP. CK represents the control group without the addition of the SARP. Data are presented as mean ± SEM from 3 independent assays. (E) Assembly of the SARP-TIC. The protein compositions in the dotted line boxed fractions are shown in the SDS-PAGE. The original gel image can be found in S1 Raw Images. (F) The *afsS* promoter fragment used for the SARP-TIC assembly. The −35 element, −10 element, the TSS, and the 6-bp noncomplementary bubble are denoted. The *afs box* is colored orange and contains DRup (the upstream direct repeat) and DRdown (the downstream direct repeat). The top (non-template, NT) strand and bottom (template, T) strand are colored light green and dark green, respectively. (G) Two views of cryo-EM map. The map was generated by merging the consensus map of the full SARP-TIC and the focused map of the SARP region in Chimera X. (H) Cartoon representation of the SARP-TIC structure. The subunits are colored as in the color scheme. The data underlying C, D, and E are provided in S1 Data. BTA, bacterial transcriptional activation; cryo-EM, cryo-electron microscopy; NOD, nucleotide-binding oligomerization domain; ODB, OmpR-type DNA-binding; RNAP, RNA polymerase; SARP, *Streptomyces* antibiotic regulatory protein; TIC, transcription initiation complex; TPR, tetratricopeptide repeat; TSS, transcription start site.

include APAF1/CED4 that regulates apoptosis [26] and plant resistance proteins that respond to pathogens and stress [27], as well as bacterial transcriptional activator MalT in *Escherichia coli* [12,28,29]. STAND proteins are presumed to keep in a monomeric inactive state by arm-based NOD–sensor autoinhibitory interactions or inhibitor binding in the absence of cognate inducer molecules [26,30–32]. Activation involves a multistep process of inducer binding, autoinhibition release, nucleotide exchange, and conformational changes that switch the protein to an active state [33]. However, the mechanisms by which AfsR regulates itself and activates transcription are still obscure.

Here, we report cryo-electron microscopy (cryo-EM) structures of the SARP-transcription initiation complex (TIC), elucidating a transcription-activating mechanism that could be generally used by SARP family activators. We find AfsR uses its SARP domain to engage the *afs box* and make extensive interactions with all RNAP subunits except ω subunit. We confirm the inhibitory effect of NOD and TPR on SARP transcription activation by in vitro assays. The inhibitory effect can be eliminated by ATP binding, but phosphorylation of AfsR counteracts the disinhibition by ATP. Overall, we present a detailed molecular view of how AfsR activates transcription.

## 2. Results

### SARP engages DNA and RNAP to activate transcription

Despite abundant genetical and biochemical studies on SARPs, the molecular basis of SARP-based transcription activation remains unknown. The isolated SARP domain of AfsR is able to bind *afs box* as evidenced by fluorescence polarization assays (Fig 1C), and mutating the upstream DR (M1) or the downstream DR (M2) attenuates this specific interaction (S1 Fig), in agreement with the previous report [9]. In vitro MangoIII-based transcription assays demonstrated the SARP domain could activate the transcription of *afsS* promoter (Fig 1D), while it has no significant effect on a control *actII-4* promoter comprising a consensus −10 and −35 element but lacking the *afs box* (S1C Fig). Consistent with the DNA binding results, when the DRs were mutated, the SARP domain lost much of its ability to activate transcription, especially with mutation at the downstream DR (S1C Fig). To investigate the molecular basis of SARP-based transcription activation, we combined the isolated SARP domain, a DNA scaffold, RbpA, CarD, and RNAP σ$^{HrdB}$-holoenzyme to assembly the initiation complex (Fig 1E). The DNA scaffold is engineered from the *afsS* promoter, consisting of a 44-bp (−50 to −7) upstream promoter double-stranded DNA (dsDNA) which contains the 22-bp *afs box* and the consensus −10 element, a 6-bp (−6 to −1) non-complementary transcription bubble, and a 15-bp (+1 to +15) downstream promoter dsDNA (Fig 1F). RbpA and CarD, 2 RNAP-binding proteins discovered in actinobacteria, can stabilize the TIC and have been found in many TIC structures of *Mycobacterium tuberculosis* [34–36]. Through in vitro transcription assays, we observed that SARP maintained its role as a transcriptional activator in the presence of RbpA and CarD (S2 Fig).

The final cryo-EM map of the SARP-TIC was reconstructed using a total of 95,223 single particles and refined to a nominal resolution of 3.35 Å, with approximately 3 Å at the center of RNAP and approximately 6 Å at the peripheral SARP (S3 Fig and S1 Table). Local refinement focused on the SARP region generated a 3.77-Å-resolution map. In the cryo-EM maps, the densities allowed unambiguous docking of 2 SARP protomers, a 61-bp promoter DNA (−46 to +15), a σ$^{HrdB}$ subunit except disordered region 1.1, 5 subunits of RNAP core (α$_2$ββ′ω), RbpA, and CarD. In addition, the cryo-EM density map shows a clear signal for an αCTD (Fig 1G). The 2 SARP protomers simultaneously engage the *afs box* and RNAP σ$^{HrdB}$-holoenzyme. SARP-TIC is an open complex containing a transcription bubble of 13 nt. The overall structure

of *S. coelicolor* RNAP in SARP-TIC closely resembles that in our recently reported *S. coelicolor* RNAPσ$^{HrdB}$-Zur-DNA structure (PDB ID: 7X75, rmsd of 0.334 Å for 3,002 aligned Cα atoms) [37], and other actinobacteria RNAP structures including *M. tuberculosis* RNAP-promoter open complex (PDB ID: 6VVY, rmsd of 1.133 Å for 2,745 aligned Cα atoms) [38] and *Mycobacterium smegmatis* RNAP TIC (PDB ID: 5VI5, rmsd of 1.048 Å for 2,621 aligned Cα atoms) [39] (S4 Fig). As shown in Fig 1H, σ$^{HrdB}$ R4 is positioned on top of 2 SARP protomers, instead of binding in the major groove of the −35 element as observed in the reported structures [37–39], suggesting a sigma adaptation mechanism like *M. smegmatis* PafBC [40]. CarD binds the β-subunit and interacts with the upstream double-stranded/single-stranded (ds/ss) junction of the transcription bubble [41], while RbpA enhances interactions between β′, σ$^{HrdB}$, and promoter DNA [42].

## A side-by-side SARP dimer interacts with *afs box*

The SARP-TIC structure elucidates the binding interactions between SARP protomers and 2 major grooves of the *afs box*, with each protomer making contacts with 1 DR (Fig 2A). The contacts span the first 8 conserved base pairs. In contrast, the last 3 base pairs do not make any contact with SARP (Fig 2B). Overall conformations of the 2 protomers are essentially the same with an overall rmsd of 0.8 Å (S5 Fig). A dimer interface of approximately 1,120 Å$^2$ is formed between the 2 SARP protomers, revealing a unique side-by-side arrangement (Fig 2C). Since SARP is monomeric in solution, the dimerization of SARP occurs upon DNA binding. As a representative example of SARP family transcription activators, the SARP folds into an N-terminal ODB domain (residues 25–111) and a C-terminal BTA domain (residues 120–270) (Fig 2D). The ODB domain consists of 3 α helices packed against 2 antiparallel β sheets. Two helices, α2 and α3, and the seven-residue connecting loop constitute an HTH DNA-binding motif, followed by a β hairpin that contacts the strand between the helices α1 and α2. The BTA domain comprises 7 α-helices, of which the first 3 ones stand on the first β sheet and the first 2 helices of the N-terminal ODB domain, burying an interfacial area of approximately 860 Å$^2$. The overall structure of SARP protomers closely resembles EmbR (PDB ID: 2FEZ, rmsd of 2.8 Å for 247 aligned Cα atoms) [6], a transcriptional regulator of *M. tuberculosis*, except that EmbR comprises an additional C-terminal forkhead-associated (FHA) domain (S6 Fig).

The 2 SARP protomers establish a total contact surface of approximately 1,300 Å$^2$ with the dsDNA, resulting in a 20˚ bend of the helical axis of the upstream dsDNA at T$_{-30}$(nt) (-30T on the non-template strand) (S7 Fig). The DNA bending at the midpoint between the 2 binding sites suggests that both protomers equally contribute to curve the DNA. The DNA contacts made by the upstream SARP protomer were described in the following section (Fig 2E). The N-terminal end of helix α1, the C-terminal end of helix α2, and the HTH loop make extensive contacts with the backbone phosphate groups from C$_{-40}$ to T$_{-37}$ of the nt-strand, involving both hydrogen bonds (S46 and Q48) and van der Waals interactions (W74, P79, S80, and Q81). The recognition helix α3 penetrates the DNA major groove and is almost perpendicular to the DNA axis. R87, R94, and K95 form salt bridges with the backbone phosphate groups from T$_{-35}$ to G$_{-33}$ of the t-strand. The side chain of S91 makes a hydrogen bond with the phosphate group of T$_{-35}$(t). In addition to these nonspecific contacts with backbone phosphate groups, the recognition helix α3 also establishes specific DNA contacts (Fig 2F). The R92 makes hydrogen bonds with the O4 atom of T$_{-38}$(nt) via its guanidinium group. The side chain of T88 makes a hydrogen bond with N7 of G$_{-36}$(t). Consistent with the structural observations, previous EMSAs show that changing the T$_{-38}$(nt) of the upstream DR or the corresponding T$_{-27}$(nt) of the downstream DR to adenosine prevents the binding of AfsR [21]. However, changing the G$_{-36}$(t) to adenosine does not impair the binding of AfsR since A$_{-25}$(t) is observed

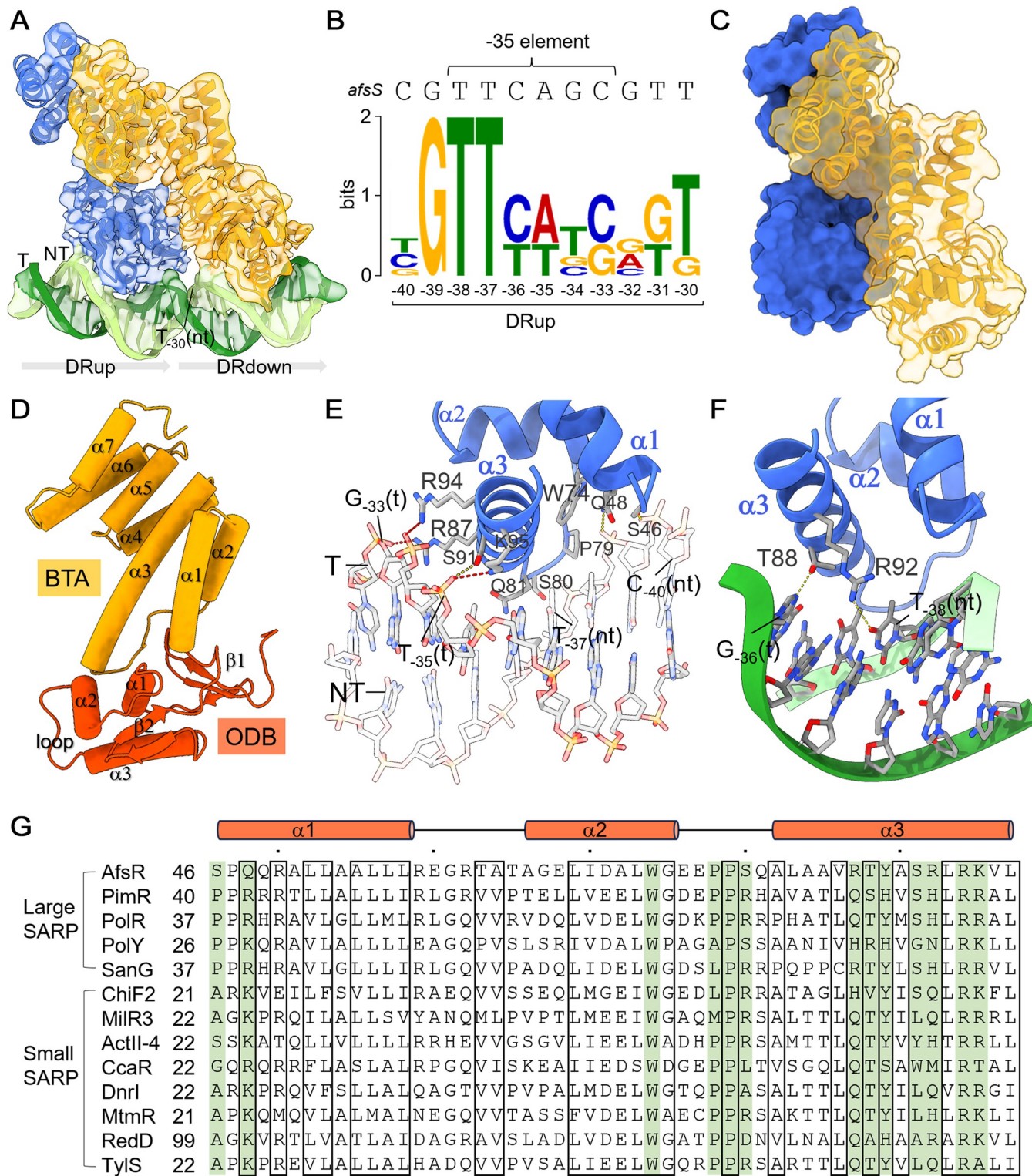

**Fig 2. Interactions of SARP with *afs box*.** (A) Two SARP protomers bind to the *afs box* (−19 to −40) side-by-side with each protomer contacting 1 DR. The density map is shown as transparent surface. T_{−30} (nt) is the last nucleotide of DRup. Upstream and downstream SARP are colored blue and orange, respectively. (B) Conserved sequences corresponding to the 11-nt DR of the *afs box* generated by MEME. (C) The dimer interface between 2 SARP protomers. (D) The domain organization indicated in the downstream SARP protomer. (E) Detailed interactions of the upstream protomer with the DNA backbone. Hydrogen bonds and salt bridges are shown as yellow and red dashed lines, respectively. (F) Contacts of SARP with specific nucleotides. The residues R92 and T88 make hydrogen bonds (shown as yellow dashed lines) with O4 of T_{−38}(nt) and N7 of G_{−36}(t), respectively. (G) Sequence alignments of SARP regulators

from different *Streptomyces* strains, highlighting the residues interacting with DNA (green). Only ODB domains were compared. These proteins include AfsR (P25941), ActII-4 (P46106), RedD (P16922) from *S. coelicolor*, PimR (Q70DY8) from *S. natalensis*, PolY (ABX24502.1), PolR (ABX24503.1) from *S. asoensis*, SanG (Q5IW77) from *S. ansochromogenes*, ChlF2 (Q0R4N4) from *S. antibioticus*, MilR3 (D7BZQ7) from *S. bingchenggensis*, CcaR (P97060) from *S. clavuligerus*, DnrI (P25047) from *S. peucetius*, MtmR (Q194R8) from *S. argillaceus*, and TlyS (M4ML56) from *S. fradiae*. The black boxes highlight the positions conserved. DR, direct repeat; ODB, OmpR-type DNA-binding; SARP, *Streptomyces* antibiotic regulatory protein.

at the corresponding position of the downstream DR [21]. The 2 ODB domains make similar contacts with DNA, inserting the helix α3 of the HTH into the major groove of the DR (S8 Fig). Most residues involved in DNA interactions are highly conserved in both "small" and "large" SARP homologues across various *Streptomyces* strains (Fig 2G), including Q48, W74, P79, T88, R94, and K95.

## SARP establishes extensive interactions with RNAP

Both SARP protomers make contacts with σ$^{HrdB}$ R4, burying a total interface of 800 Å$^2$ (Fig 3A). The upstream protomer contacts σ$^{HrdB}$ R4 by its ODB domain, of which the HTH forms a concave surface at the N-terminal of the helix α3 to enfold the N-terminal of the helix α4 of σ$^{HrdB}$ R4. The negatively charged residues E76 and E77 in the loop connecting helices α2 and α3, and the residues E68 and D71 in helix α2 of the ODB domain form salt bridges with R485 and R487 in σ$^{HrdB}$ R4, respectively (Fig 3B). The interface where the shape and charge complement each other buries an area of approximately 320 Å$^2$. The downstream protomer mainly uses its BTA domain to contact σ$^{HrdB}$ R4, with a buried surface area of approximately 480 Å$^2$ (Fig 3C). The helices α3 to α7 of the downstream BTA domain form a "chair" to let σ$^{HrdB}$ R4 sit on it. The C-terminal of helix α4, the N-terminal of helix α5, and the connecting turn of σ$^{HrdB}$ R4 fit in a groove formed by helices α3 to α7 of the BTA domain. The R498 at the C-terminal of helix α4 of σ$^{HrdB}$ R4 makes contacts with the E176 and V172 on the "chair back." The H499 in the connecting turn of σ$^{HrdB}$ R4 is enfolded in an amphiphilic pocket formed by residues L211, L243, L247, V249, E213, and R252 of the BTA domain. The K496 at the C-terminal of helix α4 of σ$^{HrdB}$ R4 makes a salt bridge with the E246 at the C-terminal of helix α6 of the BTA domain. In addition, the HTH loop of the ODB domain forms interactions with the loop connecting helices α2 and α3 of σ$^{HrdB}$ R4.

The downstream protomer also interacts with the RNAP β and β′ subunits. The ODB domain contacts the β-flap tip helix (FTH), the preceding loop (TPL), and the following loop (TFL) of the β flap by the helices α1 and β2, and the second antiparallel β sheet (Fig 3D). The residues D71 and E76 of the ODB make salt bridges with the R805 of the β FTH. The R59 at the C-terminal of the ODB α1 forms a salt bridge with the E817 in TFL. The R62 and T63 of the strand β4 connecting the helices α1 and α2 contact the R816 in TFL and the K795 in TPL, respectively. The S106 in the loop connecting the strands β5 and β6 forms a hydrogen bond with the main chain oxygen atom of the K749 in the loop following the first strand of the β flap. The BTA domain forms interactions with the C-terminal of the β FTH via the N-terminal of the helix α3, of which the R173 side chain is parallel to TFL. The positively charged R67 and R69 region of β′ zinc-binding domain (ZBD) complements the negatively charged E76 and E77 of the HTH loop of the ODB domain (Fig 3E).

To confirm the importance of the residues contacting RNAP subunits in transcription activation, they were replaced by alanine through site-directed mutagenesis. The mutants were purified and assayed by MangoIII-based transcription assays as the wild-type protein (Fig 3F). Mutating residues R59, D71, E76, and R173 diminished the capability of SARP to active transcription, whereas no obvious difference was observed between the T63A, E77A, E246A, L247A, and V249A mutants and the wild-type protein (S9 Fig). Mutating residues E176, L211,

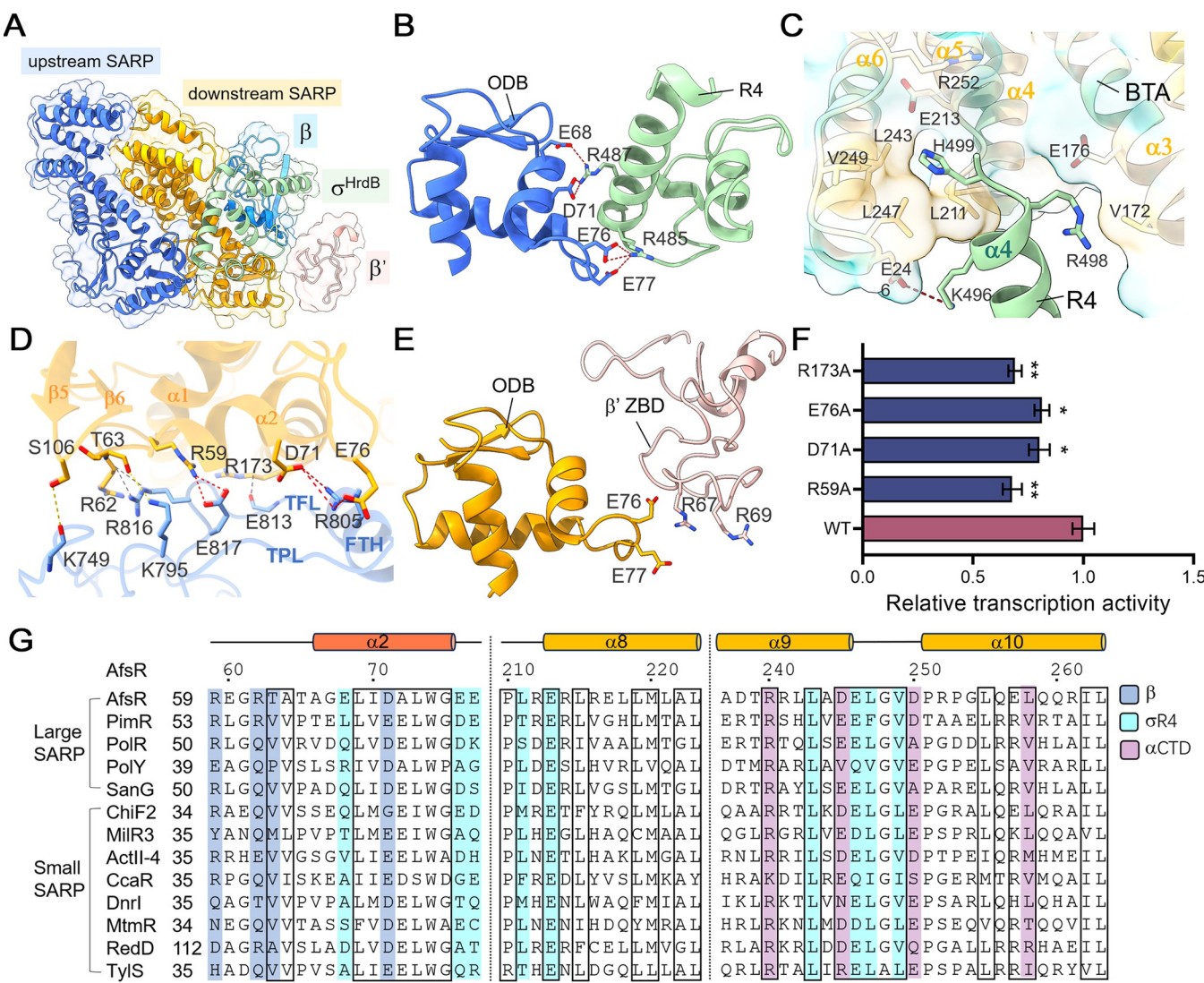

**Fig 3. SARP interacts with σ^HrdB R4, β FTH, β′ ZBD.** (A) The SARP protomers interact with σ^HrdB, β, and β′. (B) The upstream SARP protomer contacts σ^HrdB R4 by its ODB domain. Salt bridges are shown as red dashed lines. (C) The H499 of σ^HrdB R4 is enfolded in an amphiphilic pocket of the BTA domain. The K496 of σ^HrdB R4 makes a salt bridge with the E246 of the BTA domain. (D) The downstream SARP protomer make extensive interactions with the β FTH, the preceding loop (TPL) and the following loop (TFL) of the β flap. SARP is colored orange and β flap is colored blue. Hydrogen bonds, salt-bridges, and van der Waals interactions are shown as yellow, red, and gray dashed lines, respectively. (E) Interactions between the β′ ZBD and the ODB of downstream SARP. The positively charged R67 and R69 of β′ ZBD contact the negatively charged E76 and E77 of the HTH loop of the ODB domain. (F) Mutating interfacial residues of SARP impaired transcription activation. The data underlying this figure can be found in S1 Data; error bars, SEM; $n = 3$; *$P < 0.05$; **$P < 0.01$ in comparison with the wild-type SARP. (G) Sequence alignments of SARP regulators from different *Streptomyces* strains, highlighting the residues interacting with β (blue), σ R4 (cyan), and αCTD (purple). The black boxes highlight the positions conserved. BTA, bacterial transcriptional activation; FTH, flap tip helix; ODB, OmpR-type DNA-binding; SARP, *Streptomyces* antibiotic regulatory protein; ZBD, zinc-binding domain.

and L243 resulted in insoluble proteins when they were expressed under the same conditions as the wild-type protein. Some residues involved in RNAP interactions are highly conserved in SARP homologues across various *Streptomyces* strains (Fig 3G), including T63 and D71 for interaction with the β subunit, L243, E246, L247, and V249 for interaction with σ R4.

αCTD has versatility and important regulatory roles in modulating RNAP performance. In *E. coli* RNAP, it recognizes specific sequences called UP elements in some promoters [43]. Many regulators have been reported to activate transcription by interacting with αCTD. The

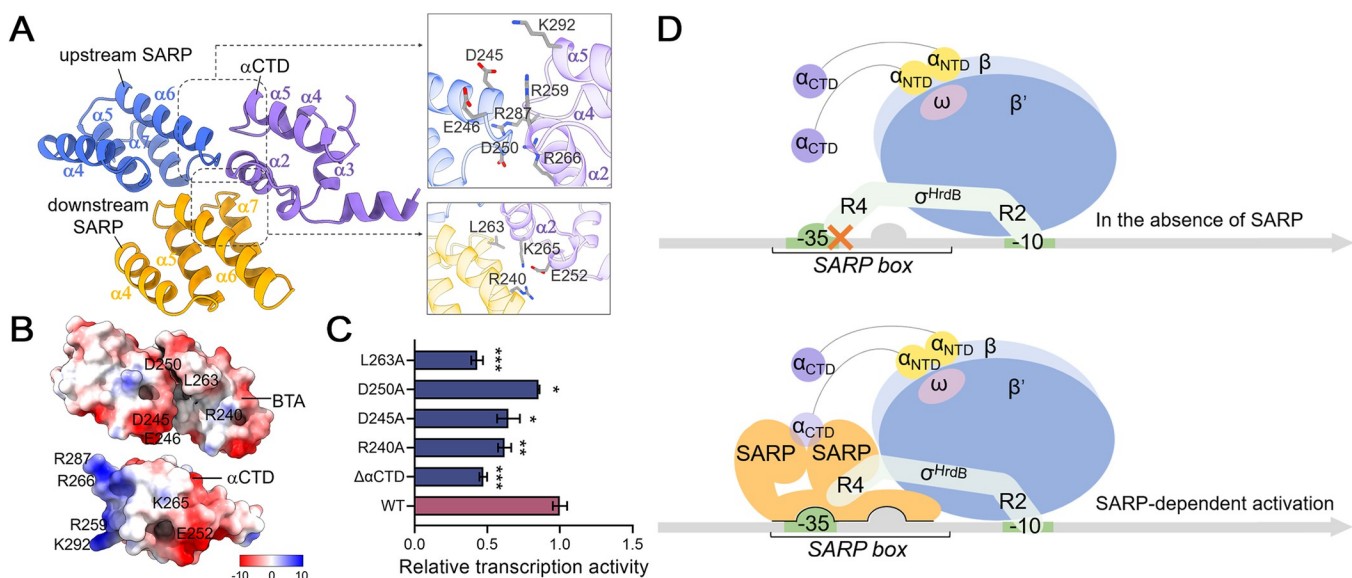

**Fig 4. Interactions of SARP with RNAP αCTD.** (A) BTA domains of both SARP protomers interact with the αCTD. Detailed interactions are shown in the gray box. (B) The electrostatic potential interface of BTA domains composed of R240, D245, E246, D250, and L263, and that of RNAP αCTD composed of E252, R259, K265, R266, R287, and K292. (C) Removing αCTD or mutating SARP residues involved in interactions with αCTD impaired transcription activation compared with wild-type SARP. The data underlying this figure can be found in S1 Data; error bars, SEM; $n = 3$; *$P < 0.05$; **$P < 0.01$; ***$P < 0.001$ in comparison with the wild-type SARP. (D) Proposed model for SARP-dependent transcription activation. BTA, bacterial transcriptional activation; RNAP, RNA polymerase; SARP, *Streptomyces* antibiotic regulatory protein.

SARP-TIC structure demonstrates that both BTA domains interact with the αCTD, burying an interface area of approximately 660 Å² (Fig 4A). The αCTD is positioned on top of the helix bundle of the downstream BTA domain. The helix α2 of αCTD interacts with the helix bundles of both upstream and downstream BTA domains and is almost perpendicular to them. The loops preceding and following helix α2, and the loop connecting helices α4 and α5 of αCTD make contacts with the helix bundle of the downstream and upstream BTA domain, respectively. The αCTD interface is predominantly positively charged (R259, R266, R287, K292), complementing the negatively charged interface of BTA domains (D245, E246, and D250 of the upstream BTA domain) (Fig 4B). The residues R240 and L263 of the downstream BTA domain interact with αCTD by electrostatic and hydrophobic interactions, respectively. As shown in Fig 4C, the αCTD is required for SARP-dependent activation. Site-directed mutational studies further confirmed the importance of interfacial residues for SARP-dependent transcriptional activation. Moreover, most residues mentioned above for contacting the αCTD are highly conserved in SARP homologues, including R240, D245, E246, and L263 (Fig 3G).

## Transcription activation of SARP is regulated by C-terminal domains

AfsR is a large SARP protein that contains additional NOD and TPR domains at its C-terminus, belonging to STAND family. STAND proteins are presumed to keep in an inactive state in the absence of stimulus signals and the activation involves a multistep of conformational changes. Here, we aimed to investigate whether the SARP effector domain of AfsR is similarly regulated. We then expressed and purified full-length AfsR and a truncation lacking the TPR domain (ΔTPR). Consistent with the isolated SARP, AfsR and ΔTPR migrate primarily as monomers in the SEC column [9] (S10 Fig). Notably, when assessing the transcription activation of full-length AfsR and ΔTPR, we observed that they exhibited lower levels compared to

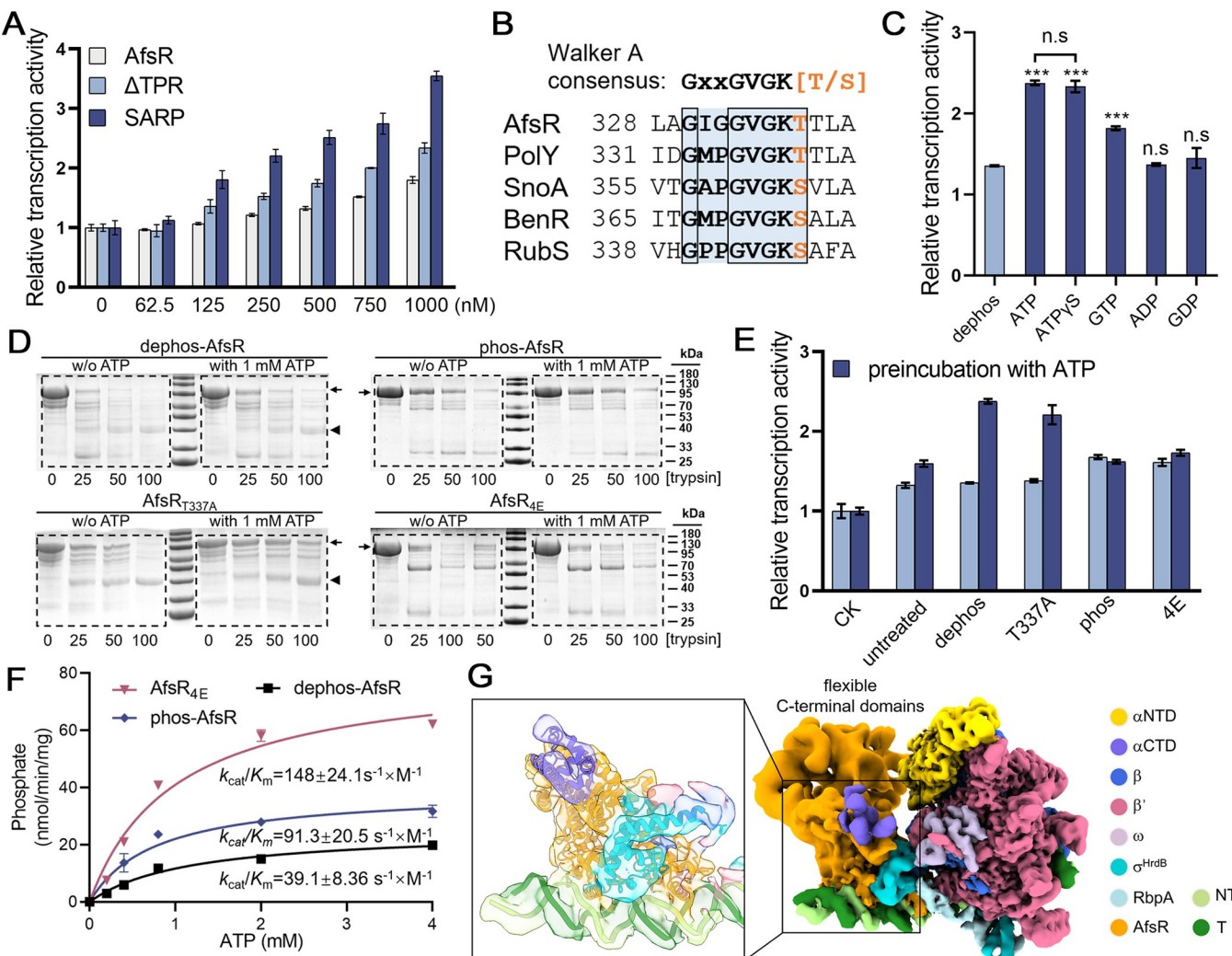

**Fig 5. ATP binding activates unphosphorylated AfsR.** (A) Transcription assays with increasing concentrations of AfsR as well as its truncations ΔTPR and SARP. (B) Sequence alignment of *Streptomyces* STAND family members highlighting the consensuses sequences of Walker A motif. The T337 phosphorylated by *E. coli* kinases is colored orange. (C) Transcription assays of 500 nM dephosphorylated AfsR preincubated with different additional nucleotides. Data are presented as mean ± SEM from 3 independent assays. (D) Representative SDS-PAGE analysis of proteolysis resistance of dephosphorylated AfsR, phosphorylated AfsR by $AfsK_{\Delta C}$, $AfsR_{T337A}$, and $AfsR_{4E}$ in the absence or presence of 1 mM ATP. Arrows and triangles indicate the primary AfsR bands and major degradation bands, respectively. Increasing concentrations of trypsin (0–100 μg/ml) were used. The original gel images can be found in S1 Raw Images. (E) Transcription assays of 500 nM AfsR (untreated), dephosphorylated AfsR (dephos), phosphorylated AfsR (phos), $AfsR_{T337A}$ and $AfsR_{4E}$ with or without preincubation with 1 mM ATP. CK represents the control group without the addition of AfsR. Data are presented as mean ± SEM from 3 independent assays. (F) ATPase activity assay of the $AfsR_{4E}$, phosphorylated AfsR, and dephosphorylated AfsR. The $AfsR_{4E}$ showed $K_m$ of 0.995 ± 0.152 mM and $k_{cat}$ of 0.147 ± 0.00841 s$^{-1}$. The phosphorylated AfsR showed $K_m$ of 0.767 ± 0.163 mM and $k_{cat}$ of 0.070 ± 0.00515 s$^{-1}$. The dephosphorylated AfsR showed $K_m$ of 1.16 ± 0.231 mM and $k_{cat}$ of 0.0453 ± 0.00356 s$^{-1}$. Data shown are the mean ± SEM for $n = 3$ experiments. (G) The cryo-EM map of AfsR-TIC. The map was generated by merging the consensus map of the full AfsR-TIC and the focused maps of the AfsR in Chimera X. Detailed interactions in the focused map focused on SARP are shown in the box. The data underlying A, C, E, and F are provided in S1 Data. cryo-EM, cryo-electron microscopy; SARP, *Streptomyces* antibiotic regulatory protein; STAND, signal transduction ATPases with numerous domains; TIC, transcription initiation complex.

the isolated SARP domain (Fig 5A), indicating that the NOD and TPR domains exert an inhibitory effect on the transcription activation of the SARP domain.

In bacterial two-component systems, phosphorylation of the response regulator usually induces conformational changes, leading to its activation. Previous reports have indicated that AfsR can undergo phosphorylation by multiple serine/threonine kinases. Considering the possibility of phosphorylation by endogenous kinases in *E. coli*, we conducted mass spectrometry

(MS) analysis on AfsR purified from *E. coli*. The MS results revealed phosphorylation at T337, the last residue of the Walker-A motif of the nucleotide-binding domain (NBD) (Fig 5B and S2 Table). We treated AfsR with λ phosphatase. The resulting dephosphorylated AfsR exhibited unaltered transcriptional activity compared to the untreated counterpart (Fig 5A and 5C). To imitate the phosphorylated AfsR in *S. coelicolor*, we purified AfsK$_{\Delta C}$, a kinase encoded by a gene located in the same cluster as the *afsR*. We utilized AfsK$_{\Delta C}$ to phosphorylate AfsR (pre-dephosphorylated) and subsequently performed MS assays. The results indicated phosphorylation at residues S22, T337, S391, T506, and S953 (S11A Fig and S2 Table). Among these phosphorylated residues, T337, S391, T506, and S953 in NOD and TPR domains are conserved (S11B Fig). Furthermore, phosphorylated AfsR showed slightly higher transcriptional activation compared to the unmodified form (S11C Fig). It has been demonstrated that in vitro phosphorylation of AfsR by AfsK yields a very small population of the phosphorylated AfsR due to the low phosphorylation efficiency [10]. To further investigate the impact of phosphorylation, we mimicked the phosphorylation by introducing a glutamate substitution at each site. Four mutations (S391E, T337E, T506E, and S953E) marginally contribute to transcriptional activation (S11D Fig). Subsequently, glutamate substitutions were introduced at all 4 sites (4E). The 4E mutant formed more oligomers (S11E Fig) and demonstrated stronger binding to the *afs box* (S11F Fig) compared to the dephosphorylated AfsR, which is consistent with previous experiments conducted with phosphorylated AfsR [10]. However, the transcription activation of the 4E mutant was still only slightly higher than that of dephosphorylated AfsR (S11D Fig), suggesting that phosphorylation mediated by AfsK likely does not trigger activation of AfsR.

The NOD domain of AfsR is known to hydrolyze ATP and GTP [10]. Thus, AfsR was preincubated with ATP or GTP to test the effects on transcription activation. As depicted in Fig 5C, preincubating dephosphorylated AfsR with ATP or GTP significantly enhances its transcription activation. In the case of incubation with ATP, the transcription activation is comparable to that of the isolated SARP (Fig 5A and 5C). However, preincubation with ADP or GDP exerted no discernible influence. Besides, the activation of AfsR was likely triggered by ATP binding rather than ATP hydrolysis, as evidenced by the comparable effect of preincubation with a non-hydrolyzable ATP analogue, ATPγS. This behavior is reminiscent of MalT, another STAND regulator, where ATP binding instead of ATP hydrolysis is essential for transcription activation [44,45]. To gain further insights, limited protease digestion experiments were conducted, revealing that preincubation with ATP induces conformational changes in dephosphorylated AfsR and AfsR$_{T337A}$ mutant so that they are more resistant to trypsin digestion (Fig 5D). However, preincubation with ATP does not affect the oligomerization state or DNA-binding capacity of AfsR (S12 Fig). Interestingly, phosphorylated AfsR and the 4E mutant, which mimics phosphorylation, are no longer activated by preincubation with ATP (Fig 5E), likely due to the increased ATPase activity (Fig 5F), which converts ATP to ADP. Phosphorylated AfsR, along with the 4E mutant, exhibits partial activation in response to non-hydrolysable ATPγS (S13 Fig). In addition, the limited protease digestion profiles of phosphorylated AfsR and the 4E mutant remained unchanged in the presence of ATP. However, the major digestion bands detected around 40 kDa in dephosphorylated AfsR were absent in phosphorylated AfsR and the 4E mutant (Fig 5D).

We tried to find out the interplay between SARP and its C-terminal domains during transcription activation by assembling AfsR-TIC (S10 Fig). AfsR$_{T337A}$ was used to eliminate the interference of *E. coli* endogenous kinases and ATPγS was used to active AfsR. The final cryo-EM map was refined into a nominal resolution of 3.63 Å (S14 Fig). But the density extending from the C-terminal of SARP was very ambiguous and could not be improved by 3D classification and local refinement, presumably due to the high flexibility of the NOD and TPR domains

relative to the SARP domain. The structures of SARP and RNAP in the AfsR-TIC resemble those in SARP-TIC (Figs 5G and S15), indicating that the SARP domain in the full-length AfsR and the isolated SARP activates transcription by similar mechanism.

## 3. Discussion

Streptomycetes exhibit a wide diversity in promoter sequences and transcription patterns to finetune the secondary metabolism to synthesize numerous natural products in response to environmental change. SARPs are well-known transcription activators of antibiotic biosynthesis. The majority of filamentous Actinobacteria genera are known to possess SARP-type regulatory genes, with examples including 98% of *Streptomyces* and 100% of *Salinispora*, while the occurrence in *Mycobacterium* is lower, at only 42% [46]. "Small" SARP regulators exhibit higher prevalence than "large" ones [7]. These "small" SARPs hold significant potential as tools for activating biosynthetic gene clusters, as evidenced by the manipulations of SARPs to enhance the yield of natural products in the native producers or the heterologous hosts [7,47–49].

The diminished basal activity of SARP-regulated promoters may be attributed to deficiencies in their −10 region, −35 region, and the extended spacer between the −10 and −35 elements. Here, we report a strategy used by SARP to circumvent the recognition of a suboptimal promoter and activate transcription. Given the comparable protein architecture of SARPs and their binding to similar motifs, SARPs are likely to function in a similar mode (Fig 4D). The position of *afs box* (centered at −29.5) is different from the binding boxes of typical class II transcriptional activators (centered at −41.5). Consequently, the downstream SARP protomer is closer to RNAP and makes extensive contacts with β and β′ subunits (Fig 3A). Within the scope of our investigation, SARP demonstrates the most comprehensive interaction with RNAP among the studied transcription activators, encompassing all subunits of RNAP except for ω, namely α, β, β′, and σ subunits. Particularly, SARP provides an alternative docking point for σ R4. In reported class II activator-TIC structures [23], the σ R4 binds in the major groove of the −35 element and interacts with class II activators such as CAP [22], SoxS [50], and Rob [51] (S16 Fig). The transcriptional activation mode of SARP mostly resembles that of the mycobacterial transcriptional activator PafBC termed "sigma adaptation" [40] (S16F Fig). PafBC inserts between PafBC-specific −26 element and σ R4 to facilitate the recognition of promoters with suboptimal −35 elements. It has been reported that most promoters of *S. coelicolor* and *M. tuberculosis* have a conserved −10 element, but a highly variable −35 element [18,19,52]. Our study showed the "sigma adaption" principle compensating for the suboptimal −35 elements may be widely adopted in actinobacteria, as indicated by the ubiquitous presence of SARP regulators. In addition, both SARP protomers make contacts with the αCTD, enhancing the ability to activate transcription by recruiting RNAP. Notably, although many activators in reported TIC structures are dimer, they only use 1 protomer to contact αCTD. In class I CAP-TAC structures, the downstream protomer contacts the αCTD [22]. In contrast, the upstream protomer contacts the αCTD in class II TAP-TAC structures [23].

Like other STAND regulators, the transcription activation of AfsR exerted by the SARP effector domain is inhibited by the NOD and TPR domains (S17 Fig). By in vitro transcriptional assays, we demonstrate that preincubation of AfsR with ATP enhances its capability of activating transcription comparable to that of the isolated SARP domain. Preincubation with ADP does not enhance transcription activation of AfsR, suggesting a role for the triphosphate molecule in modulating AfsR function. Limited protease digestions indicate ATP may induce conformational changes of AfsR. However, the precise mechanism still requires further investigation. The prototypical bacterial STAND regulators MalT and GutR are activated by the sequential binding of inducer (maltotriose and glucitol, respectively) and ATP [29,53]. AfsR

contains a tentative TPR sensor domain; however, it remains elusive whether there is an inducer binding to the AfsR sensor domain. AfsR is phosphorylated by several kinases including AfsK, which is located upstream of AfsR and separated by 2 genes. We found 5 phosphorylation sites in AfsR. The 4 phosphorylation sites (T337, S391, T506, and S953) in NOD and TPR are highly conserved in AfsR homologs (S11B Fig). In contrast to the response regulators of two-component systems that are activated by phosphorylation, phosphorylation modifications have little effect on AfsR transcriptional activation. However, preincubation of phosphorylated AfsR with ATP does not enhance its transcription activation, suggesting that phosphorylation may counteract the effects of ATP binding. We observed that phosphorylated AfsR exhibits higher ATPase activity, but further investigations are required to decipher the detailed mechanism of how ATP binding and phosphorylation modulate AfsR transcription activation.

In summary, transcription activators of the SARP family have been widely found in actinomycetes and play important roles in regulating secondary metabolism. Our detailed structural and functional analysis provides a molecular basis for understanding SARP-mediated antibiotic regulation in streptomycetes and offers potential optimizations for antibiotic production.

## 4. Materials and methods

### Plasmids

The genes encoding AfsR, ΔTPR (residues from 1 to 618 of AfsR), SARP (residues from 1 to 270 of AfsR), RbpA, and CarD were cloned from the genomic DNA of *S. coelicolor* M145, and inserted into the pET28a via *Nde*I and *EcoR*I restriction sites to obtain pET28a-*SARP*, pET28a-*rbpA*, and pET28a-*carD*, respectively. The gene encoding AfsK$_{\Delta C}$ (Met1 to Arg311) was cloned from the genomic DNA of *S. coelicolor* M145 and inserted into the pET32a vector. Plasmids carrying SARP mutants were constructed using site-directed mutagenesis. Primers used are listed in S3 Table.

### Protein expression and purification

The vector pET28a-*afsR*, pET28a-*ΔTPR*, pET28a-*SARP*, pET32a-*afsK$_{\Delta C}$*, pET28a-*rbpA*, and pET28a-*carD* were transformed into BL21(DE3), respectively. Induction of expression was achieved by adding 0.3 mM isopropyl-β-D-thiogalactopyranoside (IPTG) and incubated for 12 h at 16˚C and 220 rpm. The cells were then harvested and resuspended in buffer A (50 mM Tris (pH 8.0), 500 mM NaCl, 10% glycerol, 1 mM β-mercaptoethanol, and 5 mM imidazole). After purification by nickel-NTA, the eluate was further loaded onto a size-exclusion chromatography (SEC) column (Superdex 200, Cytiva) equilibrated with buffer B (20 mM Tris (pH 8.0), 150 mM NaCl, 5% glycerol) and the purified proteins were stored at −80˚C. Purification of *S. coelicolor* RNAP and σ$^{HrdB}$ was carried out as described previously [37].

### In vitro transcription

The transcription activities are evaluated by MangoIII-based transcription assay as previously described [37]. DNA fragments containing the *afsS* promoter (−50 to +15) or its mutants and the MangoIII sequence were used as transcription templates. Primers used are listed in S3 Table. A gradient concentration range of AfsR or its variants (ranging from 62.5 nM to 1,000 nM) was combined with 10 nM of promoter DNA at 30˚C for 10 min, followed by supplementation of 100 nM of RNAP-σ$^{HrdB}$ and 1 mM of NTPs (0.25 mM NTP each) in transcription buffer (20 mM Tris–HCl (pH 8.0), 100 mM KCl, 5 mM MgCl$_2$, 1 mM DTT, 4 U RNaseIn, 1 μm TO1–PEG–biotin, and 5% glycerol). The reactions were incubated for 15 min at 30˚C

and stopped by 0.5 ng/μl (final concentration) heparin. In the investigation of the impact of RbpA and CarD on the functionality of SARP and the transcription of *afsS* promoter, as well as subsequent experiments involving AfsR or SARP mutants, RbpA and CarD were co-incubated within the transcriptional system as global transcription factors [54,55]. A total of 500 nM (final concentration) of AfsR, SARP or their variants, RbpA, and CarD were used. For investigating the impact of ATP, ATPγS, GTP, ADP, or GDP, 500 nM dephosphorylated AfsR was preincubated with 1 mM corresponding NTP or NDP for 10 min at 30˚C and transferred into the reaction mixture (final concentration of additional nucleotides was less than 0.02 mM to avoid the influence on transcription). The reaction without NTP was used as blank. A multi-detection microplate reader (Tecan Spark) was used to measure fluorescence intensities. The emission wavelength was 535 nm and the excitation wavelength was 510 nm.

## Fluorescence polarization (FP)

A 32-bp *afs* box (−45 to −14) or its mutants consisting of a 5′-conjugated FAM was synthesized, annealed, and used as the DNA probe. The reaction mixture contained 10 nM of labeled dsDNA and 0 to 2 μm purified AfsR, SARP, or their variants in binding buffer (20 mM HEPES (pH 7.5), 100 mM NaCl, 5 mM $MgCl_2$, 2 mM DTT, 5% glycerol, and 0.1 μg/ml of poly(dI-dC)). After incubation at 30˚C for 20 min, the fluorescence polarization of the reaction mixture was detected by a multi-detection microplate reader (Tecan Spark) with excitation wavelength of 485 nm and emission wavelength of 535 nm. Data from technical triplicates were fitted to a binding equation $Y = B_{max}*X/(k_{D\text{-ave}}+X)$ to obtain the $k_{D\text{-ave}}$, where Y is the ΔmP measured at a given protein concentration (X) and $B_{max}$ is the maximum ΔmP of completely bound DNA.

## ATPase activity

ATPase activities were evaluated by using Malachite Green Phosphate Detection Kit (Beyotime). Each reaction consisted of 70 μl volumes containing 5 μm of protein in a reaction mixture of 20 mM Tris-HCl (pH 7.5), 150 mM NaCl, 5 mM $MgCl_2$, and ATP at varying concentrations. Thirty minutes after ATP addition, a multi-detection microplate reader (Tecan Spark) was used to measure the absorbance at 620 nm. At least 3 technical replicate data were fitted to the Michaelis–Menten equation to generate kinetic parameters.

## Dephosphorylation and phosphorylation of AfsR proteins

AfsR protein (5 μm) purified from *E. coli* was treated with λ protein phosphatase (100U, Beyotime) in the reaction buffer (50 mM HEPES (pH 7.5), 10 mM NaCl, 1 mM $MnCl_2$, 2 mM DTT, 0.01% Brij35) for 30 min at 30˚C, and then loaded onto the Superdex 200 SEC column equilibrated with 20 mM Tris-HCl (pH 8.0), 150 mM NaCl, and 5% glycerol to remove the phosphatase and the aggregation and to obtain dephosphorylated AfsR. In vitro phosphorylation of the pre-dephosphorylated AfsR was achieved by mixing the protein with $AfsK_{\Delta C}$ in a reaction buffer (40 mM Tris-Cl (pH 8.0), 5 mM $MgCl_2$, 5 mM $MnCl_2$, and 5% glycerol) for 30 min at 30˚C, followed by separation by the Superdex 200 SEC column equilibrated with the 20 mM Tris (pH 8.0), 150 mM NaCl, 5% glycerol to obtain the phosphorylated AfsR.

## LC-MS/MS analysis

AfsR and phosphorylated AfsR were digested with trypsin (Promega) and analyzed with Nano-LC-MS/MS system (Easy nLC1200/Q Exactive Plus, Thermo Scientific, Bremen, Germany), respectively. The mobile phase buffer A was 0.1% formic acid in water and the mobile

phase buffer B was 80% acetonitrile with 0.1% formic acid. The flow rate was 300 nL/min, and the procedure was 2% to 20% buffer B in 42 min, 20% to 35% in 5 min, 35% to 100% in 1 min, and then 100% for 12 min. For MS, the scan ranged from 350 to 1,800 m/z. For MS/MS, the scan ranged from 200 to 2,000 m/z. Sequence analysis was performed by proteome Discoverer (version 2.5) software using the sequence of AfsR, AfsK$_{\Delta C}$, trypsin as well as the database of expression strain *E. coli* BL21(DE3) and up to 2 missed cleavages were permitted. The maximum false discovery rate (FDR) for both peptides and proteins was set at 1%.

## Assembly of SARP-TIC and AfsR-TIC complex

To construct SARP-TIC and AfsR-TIC, we used a DNA scaffold engineered from *afsS* promoter carrying 6-bp pre-melted transcription bubble. The DNA was prepared by annealing nt-strand (100 μm final) and t-strand (110 μm). For assembly of SARP-TIC, 32 μm SARP was preincubated with 5 μm *afsS* promoter for 10 min at 25˚C. At the same time, 4 μm RNAP core, 16 μm σ$^{HrdB}$, 16 μm CarD, and 32 μm RbpA were incubated for 10 min at 25˚C, then mixed with preincubated SARP-DNA and incubated for another 10 min at 25˚C in assembling buffer (20 mM Tris-HCl (pH 8.0), 50 mM KCl, 5 mM MgCl$_2$, and 3 mM DTT). For assembly of AfsR$_{T337A}$-TIC, 32 μm AfsR$_{T337A}$ was preincubated with 5 μm *afsS* promoter as well as 1 mM ATPγS for 10 min at 25˚C, and 4 μm RNAP core, 16 μm σ$^{HrdB}$, 16 μm CarD, and 32 μm RbpA were incubated for 10 min at 25˚C, then mixed with preincubated AfsR$_{T337A}$-DNA and incubated for another 10 min at 25˚C in assembling buffer. We then loaded the complexes onto a Superose 6 10/300 GL column (GE Healthcare), eluted with assembling buffer, and used them directly for cryo-EM grid preparation.

## Cryo-EM data acquisition and processing

Approximately 0.5 mM ATPγS was supplemented to AfsR$_{T337A}$-TIC, and 3.5 μl of the SARP-TIC or AfsR$_{T337A}$-TIC samples were added to freshly glow-discharged Quantifoil R1.2/1.3 Au 300 mesh grids. In a Vitrobot (FEI), grids were plunge-frozen in liquid ethane after being blotted for 2 s at 16˚C with 100% chamber humidity. The grids were imaged using SerialEM on Titan Krios 300 kV microscopes with a K2 detector. The defocus ranged from −1.0 to −2.0 μm, and the magnification was ×130,000 in counting mode; 32 frames per movie were collected with a total dose of 40 e$^-$/Å$^2$. For SARP-TIC, 3,741 movies were collected, while 3,078 movies were collected for AfsR$_{T337A}$-TIC.

The SARP-TIC or AfsR$_{T337A}$-TIC data was processed using CryoSPARC suite v4.2.1. After motion correction, patch CTF estimation, manual exposure curation, and template picker using selected 2D classes from blob picker, 2 rounds of 2D classification were conducted [56]. For SARP-TIC, 603,659 particles were picked and used for ab initio reconstruction (4 classes). The selected particles (193,644 particles) were used for the second round of ab initio reconstruction (3 classes) and heterogeneous refinement. The second 3D class (95,223 particles, 49.2%) was selected and refined. Then, local resolution estimation and local filtering were conducted to give final maps. By the process of particle subtraction and masked local refinements to reserve only the signal around the SARP region, a local map with improved quality was obtained (S3 Fig and S1 Table). For AfsR$_{T337A}$-TIC, 287,126 particles were picked and used for ab initio reconstruction (3 classes). The selected particles (131,066 particles) were used for the second round of ab initio reconstruction (3 classes) and heterogeneous refinement. The first 3D class (45,208 particles, 34.5%) was selected and refined by local resolution estimation as well as local filtering to give final map. By the processes of particle subtraction and masked local refinements to reserve only the signal around the SARP region or AfsR$_{T337A}$ region, 2 local maps were obtained, respectively (S14 Fig and S1 Table).

## Cryo-EM model building and refinement

The initial models of *S. coelicolor* RNAPσ$^{HrdB}$, RbpA, and CarD were generated from PDB:7X75 [37], 5TW1 [42], and 4XLR [41], respectively. The initial atomic model of AfsR was generated by AlphaFold [57]. The model of the promoter DNA was built in Coot [58]. The models of RNAP, SARP, and DNA were fitted into the cryo-EM density maps using ChimeraX [59]. The model was refined in Coot and Phenix with secondary structure, rotamer, and Ramachandran restraints [60]. The validation was performed by MolProbity [61]. The Map versus Model FSCs was generated by Phenix (S3 and S14 Figs). The statistics of cryo-EM data processing and refinement were listed in S1 Table.

## Supporting information

**S1 Fig. In vitro assays of the *Streptomyces coelicolor* AfsR SARP.** (A) Core promoter sequences used for in vitro assays. The mutations introduced into the *afs box* are highlighted. The mutated target sites contained mutations in upstream repeat (M1), the downstream repeat (M2), or both repeats (M1M2). The *actII-4* promoter was used as a control. (B) Fluorescence polarization assays of the SARP with mutant *afs box* (M1, M2, and M1M2). Error bars represent mean ± SEM of $n$ = 3 experiments. (C) In vitro MangoIII-based transcription assays with or without 500 nM SARP in the absence of RbpA and CarD. Error bars represent mean ± SEM of $n$ = 3 experiments. The data underlying B and C can be found in S1 Data.
(TIF)

**S2 Fig. RbpA and CarD do not influence the transcriptional activation capability of SARP.** In vitro transcription assays with or without RbpA and CarD on the *afsS* promoter (left) and transcription assays with or without 500 nM SARP in the presence of RbpA and CarD (right). The data underlying this figure can be found in S1 Data; error bars, SEM; $n$ = 3.
(TIF)

**S3 Fig. Single-particle cryo-EM analysis of SARP-TIC.** (A) A motion-corrected image (scale bar: 40 nm). (B) 2D classes (scale bar: 100 Å). (C) Data processing pipeline for the dataset of SARP-TIC. The final cryo-EM map of the SARP-TIC was reconstructed using a total of 95,223 single particles and refined to a nominal resolution of 3.35 Å. Local refinement focused on the SARP region generated a 3.77-Å-resolution map. (D) Validation of cryo-EM structural models. Map vs. model FSCs was generated by Phenix (Version 1.19.2). The model-map resolution for the atomic model and cryo-EM map at FSC = 0.5 cutoff was indicated in the figure and reported in S1 Table. The data underlying this figure can be found in S1 Data.
(TIF)

**S4 Fig. Comparisons of RNAP.** (A) *S. coelicolor* SARP-TIC. (B) *S. coelicolor* RNAPσ$^{HrdB}$-Zur-DNA (PDB ID: 7X75). (C) *Mycobacterium tuberculosis* RNAP-promoter open complex (PDB ID: 6VVY). (D) *Mycobacterium smegmatis* TIC (PDB ID: 5VI5).
(TIF)

**S5 Fig. Comparison of 2 SARP protomers in SARP-TIC.** Overall conformations of the 2 protomers are essentially the same with an overall rmsd of 0.8 Å.
(TIF)

**S6 Fig. Comparison of SARP with EmbR (PDB ID: 2FEZ) comprising an additional C-terminal FHA domain.**
(TIF)

**S7 Fig. The binding of 2 SARP protomers in *afs box* results in a 20° bend of the helical axis of the dsDNA.**
(TIF)

**S8 Fig. Interactions of the downstream SARP with *afs box*.** (A) Downstream SARP residues involved in interactions with the DNA backbone. (B) Contacts of downstream SARP with specific nucleotides. The residues R92 and T88 make hydrogen bonds (shown as yellow dashed lines) with $T_{-27}$(nt) and $A_{-25}$(t), respectively.
(TIF)

**S9 Fig. In vitro assays of SARP mutants.** (A) In vitro transcription assays of SARP T63A, E77A, E246A, L247A, and V249A mutants. No obvious difference was observed between the T63A, E77A, E246A, L247A, V249A mutants, and the wild-type protein. n.s. means no significance. The data underlying this figure can be found in S1 Data; error bars, SEM; $n = 3$. (B) SDS-PAGE of E176A, L211A, and L243A mutants. Inclusion bodies were formed when these mutants were expressed under the same conditions as the wild-type protein. P represents precipitation while S represents supernatant. The original gel images can be found in S1 Raw Images.
(TIF)

**S10 Fig. Purification and assembly of protein complexes.** (A) Assembly of *S. coelicolor* RNAP-$\sigma^{HrdB}$. The peak of RNAP-$\sigma^{HrdB}$ was analyzed by SDS-PAGE. (B) Purification of AfsR. (C) Purification of ΔTPR. (D) Purification of SARP. (E) Purification of RbpA. (F) Purification of CarD. (G) Purification of $AfsK_{\Delta C}$. (H) Assembly of $AfsR_{T337A}$-TIC in the presence of 1 mM ATPγS. The protein compositions in the dotted line boxed fractions are shown in the SDS-PAGE. The original gel images can be found in S1 Raw Images.
(TIF)

**S11 Fig. Phosphorylation modulates AfsR transcriptional activation.** (A) LC-MS/MS analysis showing that S22, T337, S391, T506, and S953 are phosphorylated in AfsR by $AfsK_{\Delta C}$. The lowercase letter in the peptide sequence indicates phosphorylated residue. "b" and "y" denote peptide fragment ions retaining charges at the N and C terminus, respectively. The subscript numbers indicate their positions in the identified peptide. (B) Sequence alignment of *Streptomyces* AfsR family members highlighting the consensuses sequences neighboring phosphorylation sites S22, T337, S391, T506, and S953. Orange boxes represent phosphorylation sites. (C) Transcription assays with increasing concentrations of phosphorylated AfsR by $AfsK_{\Delta C}$ (phos-AfsR) and untreated AfsR. (D) Transcription assays of 500 nM AfsR mutants mimicking dephosphorylation (T337A) and phosphorylation (S22E, T337E, T506E, S953E, and S391E/T337E/T506E/S953E (4E)). Data are presented as mean ± SEM from 3 independent assays. n.s. means no significance; *$P < 0.05$; **$P < 0.01$ in comparison with the wild-type AfsR. (E) Representative SEC assay of $AfsR_{T337A}$ and $AfsR_{4E}$. (F) Fluorescence polarization assay of $AfsR_{4E}$ and dephosphorylated AfsR (dephos-AfsR) with *afs box*. The concentration of *afs box* was 10 nM. Error bars represent mean ± SEM of $n = 3$ experiments. The data underlying A, C, D, E, and F are provided in S1 Data.
(TIF)

**S12 Fig. Oligomerization state and DNA-binding capacity of dephosphorylated AfsR are not affected by ATP binding.** (A) SEC assay of dephosphorylated AfsR in the presence (solid line) or absence (dashed line) of 1 mM ATP. The data underlying this figure can be found in S1 Data. (B) Fluorescence polarization assay of dephosphorylated AfsR with *afs box* DNA in the presence or absence of 1 mM ATP. The concentration of *afs box* DNA was 10 nM. The

data underlying this figure can be found in S1 Data; error bars, SEM; $n = 3$.
(TIF)

**S13 Fig. ATPγS activates phosphorylated AfsR and AfsR$_{4E}$.** Transcription assays involving 500 nM phosphorylated AfsR (phos-AfsR) or AfsR$_{4E}$, with and without preincubation with 1 mM ATP or ATPγS. CK represents the control group without the addition of AfsR. The data underlying this figure can be found in S1 Data; error bars, SEM; $n = 3$.
(TIF)

**S14 Fig. Single-particle cryo-EM analysis of AfsR$_{T337A}$-TIC.** (A) Motion-corrected images (scale bar: 50 nm). (B) 2D classes (scale bar: 100 Å). (C) Data processing pipeline for the data-set of AfsR$_{T337A}$-TAC. The final cryo-EM map of the AfsR$_{T337A}$-TIC was reconstructed using a total of 45,208 single particles and refined to a nominal resolution of 3.63 Å. Local refinement focused on the SARP region generated a 5.10-Å-resolution map. Local refinement focused on the AfsR region generated a 8.61-Å-resolution map. (D) Validation of cryo-EM structural models. Map vs. model FSCs was generated by Phenix (Version 1.19.2). The model-map resolution for the atomic model and cryo-EM map at FSC = 0.5 cutoff was indicated in the figure and reported in S1 Table. The data underlying this figure can be found in S1 Data.
(TIF)

**S15 Fig. Two views of cryo-EM map and structural model of AfsR$_{T337A}$-TIC.** The map was generated by merging the consensus map of the full AfsR -TIC and the focused maps of the AfsR. The initial model of RNAP and AfsR was generated from SARP-TIC and AlphaFold, respectively, and fitted into the map using ChimeraX.
(TIF)

**S16 Fig. Interactions of regulators with σR4 and −35 element.** (A) SARP. (B) Class I CAP-transcription activation complex (TAC) (PDB ID:6B6H). (C) Class II TAP-TAC (PDB ID: 5I2D). (D) SoxS (PDB ID: 7W5W). (E) RobR (PDB ID: 7VWZ). (F) PafBC (PDB ID: 7P5X). The key residues Arg and Glu supposed to contact DNA are shown as sticks.
(TIF)

**S17 Fig. Scheme model summarizing the ATP-dependent and phosphorylation-dependent regulation of AfsR.** Unmodified full-length AfsR is autoinhibited by its C-terminal domains. AfsR is activated by ATP binding and significantly stimulates the production of transcripts. The phosphorylation of AfsR results in increased ATPase activity, potentially counteracting the effects of ATP binding. Phosphorylated AfsR forms more oligomers and shows slightly higher transcriptional activation compared to the unmodified form.
(TIF)

**S1 Table. Cryo-EM data collection, refinement, and validation statistics.**
(XLSX)

**S2 Table. Phosphorylated peptides in AfsR by mass spectrometric.**
(XLSX)

**S3 Table. List of primer sequences used in this study.**
(XLSX)

**S1 Data. Numerical values used to generate graphs.**
(XLSX)

**S1 Raw Images. The original gel images contained in the figures.**
(PDF)

## Acknowledgments

We thank Jing Liu, Xinqiu Guo, and Mengyu Yan at the Instrument Analysis Center (IAC) of Shanghai Jiao Tong University for supporting the cryo-EM data collection.

## Author Contributions

**Conceptualization:** Jianting Zheng.

**Data curation:** Yiqun Wang, Xu Yang.

**Formal analysis:** Yiqun Wang.

**Funding acquisition:** Jianting Zheng.

**Investigation:** Yiqun Wang, Xu Yang, Feng Yu.

**Methodology:** Yiqun Wang.

**Project administration:** Yiqun Wang.

**Software:** Xu Yang, Jianting Zheng.

**Supervision:** Zixin Deng, Shuangjun Lin, Jianting Zheng.

**Validation:** Yiqun Wang.

**Visualization:** Yiqun Wang.

**Writing – original draft:** Yiqun Wang, Jianting Zheng.

**Writing – review & editing:** Yiqun Wang, Jianting Zheng.

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
