## [Editor Report · Decision Letter 0]

29 Sep 2023

Dear Dr Zheng, 

Thank you for submitting your manuscript entitled "Structural and functional dissection of AfsR, a transcription activator of Streptomyces antibiotic regulatory protein family" for consideration as a Research Article by PLOS Biology.

Your manuscript has now been evaluated by the PLOS Biology editorial staff, as well as by an academic editor with relevant expertise, and I am writing to let you know that we would like to send your submission out for external peer review.

Once your full submission is complete, your paper will undergo a series of checks in preparation for peer review. After your manuscript has passed the checks it will be sent out for review. To provide the metadata for your submission, please Login to Editorial Manager (https://www.editorialmanager.com/pbiology) within two working days, i.e. by Oct 01 2023 11:59PM.

Kind regards,

Richard

Richard Hodge, PhD

rhodge@plos.org

PLOS

---

## [Decision Letter · Decision Letter 1]

11 Dec 2023

Dear Dr Zheng,

Thank you for your patience while your manuscript "Structural and functional dissection of AfsR, a transcription activator of Streptomyces antibiotic regulatory protein family" went through peer-review at PLOS Biology. Please accept my sincere apologies for the long delays that you have experienced during the peer review process. Your manuscript has now been evaluated by the PLOS Biology editors, an Academic Editor with relevant expertise, and by two independent reviewers.

In light of the reviews, which you will find at the end of this email, we are pleased to offer you the opportunity to address the comments from the reviewers in a revision that we anticipate should not take you very long. Reviewer #2 notes that a control assay for the in vitro transcription assays should be provided to show that the effects are specific to transcription initiation at the afsS promoter. We will then assess your revised manuscript and your response to the reviewers' comments with our Academic Editor aiming to avoid further rounds of peer-review, although might need to consult with the reviewers, depending on the nature of the revisions.

**IMPORTANT - SUBMITTING YOUR REVISION**

*Resubmission Checklist*

*Published Peer Review*

*PLOS Data Policy*

*Blot and Gel Data Policy*

Sincerely,

Richard

Richard Hodge, PhD

rhodge@plos.org

REVIEWS:

Reviewer #1: In this paper, Wang and Zheng et. al. characterized the structure and mechanism of AfsR stimulated transcription. They determined high-resolution structure of Sarp domain of AfsR as well as full-length AfsR bound to transcription initiation complex (TIC), which together with biochemical assays illustrated how AfsR interacts with and stimulates transcription. In addition, biochemical investigation suggested a multi-layer regulation scheme of AfsR, which are duel controlled by ATP binding and phosphorylation. The paper presents significant new insights to Sarp-domain mediated transcription activation and could be general interests to broad audience. I only have several minor suggestions.

Minor:

1. Scale bars are needed for the 2D classification images in Fig S2&10,.

2. Local densities of SARP domain and C-terminal domains of AfsR, possibly in FigS2&10 or in additional figures, are needed to show the quality of the maps, especially at the DNA binding region and C-terminal domains.

3. A minor suggestion, possibly color the protein complexes and DNAs with different color in Fig S3 will be helpful to better illustrate the architecture of TICs. 

4. In the TIC structure with SARP domain (Fig 2), the author claims the two protomer are nearly identical. A structural comparison will be helpful. In addition, protomer 1 has sequence specific DNA interactions and the promotor. How about protomer 2? Does it help with sequence recognition and transcription activation? In addition, do both protomers exert similar degree of DNA bending? 

5. A scheme model to summarize the ATP dependent and phosphorylation dependent regulation of AfsR will be helpful.

Reviewer #2: This manuscript reveals some key aspects of how AfsR, a member of the Streptomyces antibiotic regulatory protein family binds at a target promoter to activate transcription, including some very interesting and novel contacts with various subunits of RNA polymerase. The work is of particular interest to anyone working with the regulation and expression of antibiotic biosynthetic clusters since AfsR is related to many key activators, and the structural model presented is likely to be broadly applicable to this family. The works is also of wider interest to the transcription field, showing another example of how flexibility in key RNAP subunits alpha-CTD and sigma can accommodate varied scaffolds that lead to activation. The work does not explore in any detail mechanistic questions of activation and the SARP role in the sequential process of initiation, but this is beyond the scope. Overall the presentation is very good and it reads very well. I have a few specific comments.

1. The CACTGT -10 is close but not consensus since a T at the first position is most common. In fact it is argued in the paper that SARPs can compensate for poor -35 elements, however since most promoters have poor -35 elements, it is just as valid to suggest that at specific SARP-regulated promoters, the promoters lack activity in the absence of the SARP due to other "poor" promoter elements e.g. the -10 element.

2. Can the authors show that the in vitro transcription assays used are specific to transcription initiation at the afsS promoter rather than an increase in general activity of the RNAP as seen with end-to-end transcription on the DNA template. The assay with a control promoter would be useful here.

3. Line 106. I think the authors mean disinhibition by ATP>

4. Line 114. The Fig 1 C/D are mislabelled.

5. Line 115. It should be made clear that thre SARP domain was used to build the TIC.

6. Line 123. In the context of activation in Streptomyces it is important to know whether the SARP can activate transcription in the absence of RbpA and CarD. It seems an obvious omission and the paper would benefit from the inclusion.

7. The DNA scaffold used to form the TIC includes dsDNA around the -10 region, so it would be important to emphasise that the DNA is melted in the final TIC - ie that the complex has isomerized to the open complex. 

8. Line 292 and associated figure. The effect of ATP on limited proteolysis patterns of dephosphorylated AfsR vs phosphorylated AfsR is not that convincing and key differences could be highlighted. From my interpretation, the bigger difference appears to be between dephos-AfsR vs phos-AfsR rather than the effect of ATP.

9. Line 295 It is proposed that ATP hydrolysis might explain why the phosphorylated AfsR is not activated by ATP binding - what happens when a non-hydrolysable analogue of ATP is used? 

10. Line 331 The phrase that SARP functions as an adaptor to physically bond the -35 and sigma R4 is a bit ambiguous and might suggest that sigma makes normal contacts with -35 that are held in place by SARP. In fact the SARP provides an alternative docking point for R4 meaning that a -35 region is not required.

---

## [Editor Report · Decision Letter 2]

22 Jan 2024

Dear Dr Zheng,

Thank you for your patience while we considered your revised manuscript "Structural and functional dissection of AfsR, a transcription activator of Streptomyces antibiotic regulatory protein family" for publication as a Research Article at PLOS Biology. This revised version of your manuscript has been evaluated by the PLOS Biology editors and the Academic Editor.

Based on our Academic Editor's assessment of your revision, we are likely to accept this manuscript for publication, provided you satisfactorily address the following data and other policy-related requests.

IMPORTANT - please attend to the following:

a) Please change your Title to the following: "Structural and functional characterization of AfsR, a SARP family transcriptional activator of antibiotic biosynthesis in Streptomyces"

b) Please address my Data Policy requests below; specifically, we need you to supply the numerical values underlying Figs 1CDE, 3F, 4C, 5ACEF, S1BC, S2, S3D, S9A, S11ACDEF, S12AB, S13, S14D, either as a supplementary data file or as a permanent DOI’d deposition. I note that you already have a supplementary data file, "SourceData.xlsx," which seems to include some of this data. Please clarify this, and include any additional data required to support the Figures.

c) Please cite the location of the data clearly in all relevant main and supplementary Figure legends, e.g. re-name "SourceData.xlsx" as "S1_Data/xlsx" and write “The data underlying this Figure can be found in S1 Data” in each relevant Figure legend.

d) Please make any custom code available, either as a supplementary file or as part of an online deposition.

We expect to receive your revised manuscript within two weeks. 

*Published Peer Review History*

*Press*

Sincerely,

Roli Roberts

Roland G Roberts PhD

Senior Editor

rroberts@plos.org

PLOS Biology

on behalf of

Richard Hodge, 

Senior Editor,

rhodge@plos.org,

PLOS Biology

DATA POLICY:

Regardless of the method selected, please ensure that you provide the individual numerical values that underlie the summary data displayed in the following figure panels as they are essential for readers to assess your analysis and to reproduce it: Figs 1CDE, 3F, 4C, 5ACEF, S1BC, S2, S3D, S9A, S11ACDEF, S12AB, S13, S14D. NOTE: the numerical data provided should include all replicates AND the way in which the plotted mean and errors were derived (it should not present only the mean/average values).

CODE POLICY

Per journal policy, as the code that you have generated is important to support the conclusions of your manuscript, we require that you make it available without restrictions upon publication. Please ensure that the code is sufficiently well documented and reusable, and that your Data Statement in the Editorial Manager submission system accurately describes where your code can be found.

DATA NOT SHOWN?

---

## [Editor Report · Decision Letter 3]

29 Jan 2024

Dear Dr Zheng,

Thank you for the submission of your revised Research Article "Structural and functional characterization of AfsR, a SARP family transcriptional activator of antibiotic biosynthesis in Streptomyces" for publication in PLOS Biology. On behalf of my colleagues and the Academic Editor, Yunsun Nam, I'm pleased to say that we can in principle accept your manuscript for publication, provided you address any remaining formatting and reporting issues. These will be detailed in an email you should receive within 2-3 business days from our colleagues in the journal operations team; no action is required from you until then. Please note that we will not be able to formally accept your manuscript and schedule it for publication until you have completed any requested changes.

Sincerely, 

Roli Roberts

Roland Roberts PhD 

Senior Editor

PLOS Biology

rroberts@plos.org

on behalf of

Richard Hodge, PhD, 

Senior Editor

PLOS Biology

rhodge@plos.org